# Glycemic Index (GI) or Glycemic Load (GL) and Dietary Interventions for Optimizing Postprandial Hyperglycemia in Patients with T2 Diabetes: A Review

**DOI:** 10.3390/nu12061561

**Published:** 2020-05-27

**Authors:** Dionysios Vlachos, Sofia Malisova, Fedon A. Lindberg, Georgia Karaniki

**Affiliations:** 1Independent Researcher, General Physician, 16451 Athens, Greece; 2Independent Researcher, Clinical Dietitian, 11142 Athens, Greece; malisovasofia@gmail.com; 3Fedon Health Clinic, N-0207 Oslo, Norway; fedon.lindberg@drlindbergs.no; 4Independent Researcher, Clinical Dietitian, 16451 Athens, Greece

**Keywords:** postprandial hyperglycemia, PPG, glycemic load, GL, glycemic index, GI, type 2 diabetes, dietary strategies, diabetes management

## Abstract

The increasing prevalence of type 2 diabetes (T2D) worldwide calls for effective approaches to its management. Strategies for diabetes have generally focused on optimizing overall glycemic control as assessed by glycated hemoglobin (HbA1c) and fasting plasma glucose (FPG) values. However, since 2001, the American Diabetes Association has established postprandial glucose (PPG) as an independent contributor to both HbA1c and diabetes complications, and increasing evidence suggests that all three glycemic parameters of HbA1c, FPG, and postprandial glucose (PPG) are independently important. Objectives: The objective of this review was to comprehensively summarize the literature on the effects of nutritional strategies incorporating glycemic index (GI)/glycemic load (GL) on the postprandial hyperglycemia in people with T2D, as well as to provide recommendations for effective dietary strategies addressing both the dietary glycemic index and load in clinical practice. Design: An advanced Pubmed search was conducted. A total of 10 randomized controlled studies met the inclusion criteria. Six studies compared low-GI with higher GI meals, three included studies that compared reduced carbohydrate content with higher carbohydrate content, and one study compared meals of low-GI (with high or low fiber) with meals of higher GI (with high or low fiber). Results: Most of the clinical trials resulted in significant improvement (*p* < 0.05) of postprandial hyperglycemia. Conclusions: Either reducing the amount of carbohydrate in a meal or increasing consumption of soluble fiber has a favorable effect on postprandial glucose excursions.

## 1. Introduction

The growing incidence of diabetes mellitus and its impact on mortality and morbidity have become global problems [1]. In 2014, almost 422 million adults across the world were reported to have diabetes mellitus (DM) and the prevalence has globally risen from 4.7% in 1980 to 8.5% [2]. In 2016, it was estimated that DM was the seventh leading cause of mortality, with approximately 1.6 million deaths directly attributed to the disease [3].

Type 2 diabetes (T2D) accounts for about 90% of all diabetes cases [4] and is characterized by hyperglycemia and insulin resistance, which are central factors in the pathogenesis of diabetic complications. More specifically, postprandial hyperglycemia appears to play a crucial role in the pathophysiology of late diabetes complications and especially in the development of cardiovascular disease (CVD). The incidence of CVD among patients with T2D is reported as being high in many studies, and the postprandial “hyperglycemic spikes” appear to be directly related to the development and progression of CVD in T2D [5]. Achieving good glycemic control is of utmost importance for patients with T2D because effective treatment of postprandial hyperglycemia results in greater CVD benefits [6] and contributes significantly in the reduction of mortality [7]. 

Nutrition is central in the management of T2D through its clear effect on weight and metabolic control. Medical nutrition therapy offers an evidence-based approach to the management of diabetes through lifestyle modifications, which are strongly recommended to be implemented prior to the commencement of pharmaceutical therapy. This is unfortunately frequently not followed by most physicians, even though antidiabetic medications are ineffective in preventing the progression of the disease, and it is estimated that almost half of the T2D patients will start insulin therapy within 10 years of diagnosis [8]. For this reason, it is crucial to identify and implement effective dietary strategies to treat hyperglycemia and prevent progression of T2D through diet and lifestyle modifications. 

There is a consensus that the quality and quantity of carbohydrates are the main predictors of glycemic response [9,10], and low glycemic index and/or low glycemic load dietary patterns have been shown to improve glycemic control in patients with T2D. The glycemic index (GI) of a food rich in carbohydrates provides an estimation of how quickly carbohydrates break down during digestion and how rapidly they are absorbed into the bloodstream [11]. Carbohydrate-rich foods that are rapidly broken down and absorbed into the bloodstream are categorized as high-GI foods. High-GI foods lead to a rapid increase of blood glucose and insulin responses following food ingestion. On the contrary, low-GI foods have a slower and smaller effect on postprandial blood glucose levels and insulin response because they are slowly digested. Several factors determine the GI of a food [12], including the type of carbohydrate (amylopectin-rich starch is readily digested and absorbed, whereas amylose-rich starch is more slowly absorbed), as well as the content of protein, fat, and quantity and type of fiber (water-soluble dietary fiber delays postprandial glycemia, whereas non-water-soluble fiber does not have this effect), and finally the food particle size and pH (lower pH food and drinks delay postprandial glycemia). The GI of food is measured in vivo on the basis of the area under the curve of postprandial glucose after ingestion of a carbohydrate-rich food containing 50 g digestible carbohydrate, compared with the same curve after ingestion of 50 g glucose. GI is then expressed as the percentage of this ratio [13]. Glycemic index does not provide information on how increased and prolonged glycemia will be when ingesting a specific amount of a carbohydrate-rich food. A separate measure called the glycemic load (GL) does both, providing a more accurate picture of a food’s real-life impact on postprandial glycemia. Watermelon, for example, has a high glycemic index (74). However, a 100 g serving of watermelon has so little carbohydrate that its glycemic load is only 4. The term low glycemic load (GL) integrates (a) the GI of the food or diet with (b) the amount of carbohydrates in a given quantity of a food, meal, or diet. A comprehensive list of the glycemic index and glycemic load for more than 1000 foods has been published [14].

According to a recent review of the effect of GI on glycemia in patients with T2D [1], dietary management strategies aiming to improve overall glycemic control and promote weight loss in patients with T2D can depend on the use of low-GI diets. However, the outcome measures of the review were glycated hemoglobin (HbA1C) and fasting blood glucose (FBG), and not postprandial glucose. Although glycated hemoglobin (HbA1C) is used to provide a representation of the average blood glucose levels over the preceding 3 months, postprandial glucose (PPG) is used as a measure to estimate levels of blood glucose 2 hours from the initiation of a meal [15]. This is important because PPG is an independent risk factor for T2D complications [5,16]. 

Many factors can influence postprandial glucose (PPG): the GI of different foods combined in a meal; the carbohydrate content; the size of a meal; the presence and the percentage of the other three macronutrients (fat, protein, and amount and type of dietary fiber) in a meal; and also factors such as hormonal secretion, gastric emptying and the sequence of all macronutrients being ingested, and meal timing [15,17].

Strategies for managing diabetes have generally focused on optimizing overall glycemic control as assessed by glycated hemoglobin (HbA1c) and fasting plasma glucose (FPG) values. However, since 2001, the American Diabetes Association [15] has established postprandial glucose (PPG) as an independent contributor to both HbA1c and diabetes complications, and increasing evidence suggests that all three glycemic parameters of HbA1c, FPG, and postprandial glucose (PPG) are independently important. Increased PPG is also followed by a rise in postprandial insulin in people who do not have diabetes. In people with type 2 diabetes, the pancreas can be sluggish about secreting insulin in response to a meal, leading to postprandial hyperglycemia [18].

The case is made for not focusing only on PPG but also of on employing the postprandial insulin assay as a more efficient tool to diagnose prediabetes and diabetes sooner than the current standards. By the time people are diagnosed using standard glucose testing, these individuals will likely have lost up to 50% of their beta cells. When a person is diagnosed with prediabetes (using standard glucose testing), the patient will have likely lost 25% of their beta cells.

The degree of postprandial insulin increase is measured by the insulin index. The insulin index of a food represents the elevation of the insulin concentration in the blood during the 2 h period after the food is ingested. The insulin index represents a comparison of food portions with equal overall caloric content (250 kcal or 1000 kJ). The insulin index can provide additional information to the glycemic index or the glycemic load because certain foods, such as lean meats, cause an insulin response even though they contain very low amounts of carbohydrates. Holt et al. [19] have noted that the glucose and insulin scores of most foods are highly correlated, but high-protein foods and bakery products that are rich in fat and refined carbohydrates “elicit insulin responses that were disproportionately higher than their glycemic responses”. They also conclude that insulin indices may be useful for dietary management and avoidance of non-insulin-dependent diabetes mellitus and hyperlipidemia.

This review, however, provides an overall insight of the potential dietary interventions that improve postprandial glucose control in patients with T2D, given that the effective and consistent control of postprandial hyperglycemia remains one of the greatest challenges and unmet needs in diabetes management.

## 2. Methods

### 2.1. Search Method 

This review includes randomized controlled trials in adults with T2D, and the outcome of interest is postprandial glycemia. A Pubmed advanced search was performed in January 2020 using the words “glycemic” (medical subject heading, MeSH terms or title/abstract) or “glycaemic” (title/abstract) in the following combinations: “glycemic” (or glycaemic), “glycemic (or glycaemic) index”, “glycemic (or glycaemic) load”, “diabetes”, and “postprandial glucose” (MeSH terms or title/abstract), applying limitations (filters) for article types (clinical trials or randomized controlled trials), date of publication (from 2009 to 2019), species (humans, MeSH terms), ages (adult +19, MeSH terms), and language (English). Only primary research on randomized control trials (RCTs) were included in this review.

### 2.2. Selection of Studies

The first round of evaluation based on titles (*n* = 605) was conducted with the aim of identifying research studies that met the inclusion criteria and excluding those that were irrelevant with regards to the content or the type of publication such as editorials, reviews, and single case study reports. For studies on populations that involved patients with either gestational and/or type 1 diabetes, we excluded any that involved ketogenic diets. In the second round of evaluation, 167 abstracts were reviewed.

Randomized controlled studies were selected on the basis of the population, intervention, comparator, and outcomes according to the PICO framework (Table 1). Full text screening of 54 eligible articles was conducted independently by two members of the review team, and when disagreements occurred, they were settled by a third reviewer. It should be noted that clinical trials incorporating reduced carbohydrate (CHO) interventions were included on the grounds that any reduction in percentage of energy intake resulting from a decrease in the carbohydrate intake will lead to a lower GL [20]. 

## 3. Results

Ten articles met the inclusion criteria presented in Table 2. These included eight crossover clinical trials and two parallel design trials. The length of the studies ranged from 5 days to 22 months. In terms of the interventions, six studies compared low-GI with higher GI meals, three included studies conducted with comparisons of reduced carbohydrate content with higher carbohydrate content, and one study compared meals of low-GI (with high or low fiber) with meals of higher GI (with high or low fiber). 

Most of the studies presented in this review resulted in significant improvement (*p* < 0.05) of postprandial hyperglycemia, as presented in Table 3. Wolever et al. [21] compared five different possible combinations of diets/meals ((1) high-GI diet and high-GI meals, (2) low-GI diet and high-GI meals, (3) low-GI diet and low-GI meals, (4) low-CHO diet and high-GI meals, (5) low-CHO diet and low-CHO meals) and showed that both the quantity and the type of carbohydrates in the diet may result in a lower postprandial glucose response in patients with T2D. It is worth mentioning that individuals who consumed a low-GI diet with low-GI meals had lower postprandial glucose than when they consumed a low-GI diet with high-GI meals. This was attributed to the potential acute beneficial impact of low-GI foods of the meal. The significant decrease in postprandial glucose incremental area under the curve (iAUC)after breakfast was reported only in those who consumed low-GI meals. Patients who consumed a low-GI diet but had a high-GI breakfast had similar glucose responses after this meal to those who consumed a high-GI diet.

This is in line with a previous study by Nisak et al. [22], where authors concluded that altering the type of carbohydrate in a meal can result in significant improvements of postprandial glycemia in patients with T2D. In that study, researchers tested two different breakfasts with respect to the type of carbohydrates. The composition of meals were planned to provide the same energy and macronutrients with the difference being that one breakfast would consist of higher GI foods (banana and white bread), resulting in a high-GI breakfast, and the second would consist of lower GI foods (red apple and wholegrain bread), resulting in a low-GI breakfast. The low-GI breakfast resulted in a significantly lower postprandial glucose glycemia in patients with T2D when compared to the high-GI that raised postprandial glucose response by 39%. However, replacing high-GI foods with low-GI foods affected macronutrient composition of the meal, increasing fiber content and reducing energy intake from carbohydrates by 10%. 

Reis et al. [23] also investigated the effect of low-GI meals on acute glycemic control by altering the type of carbohydrates of meals. They conducted a crossover clinical trial where individuals consumed either a low-GI diet (CHO per meal 43.6 ± 25.2 g. or a high-GI diet (CHO per meal 61.1 ± 36.2 g.). Significant difference in acute glycemic control between the two groups was apparent only on the first day of the dietary intervention, possibly because participants on the high-GI diet self-restricted their carbohydrate intake after the second day of the trial. This is possibly attributable to the fact that patients knew their post-meal glucose responses, as they were instructed and trained to take their own glucose measurements, and thus they became very cautious after the first day’s high values. Regarding postprandial glycemia, no correlation was found between the amount of carbohydrate consumed and participants’ glycemic control. However, researchers concluded that educating patients to adhere to a low-GI diet by advising them to replace high-GI foods with lower GI foods is an effective nutritional strategy to improve glycemic control, as exchanging high-GI foods with low-GI foods seems to be accompanied by a significant reduction in the carbohydrate content of the diet [23]. It is worth mentioning that low-carbohydrate diets have gained popularity due to their significant positive impact on weight management and glycemic control [8].

Four studies included in the analysis compared reduced carbohydrate content with higher carbohydrate content meals and showed that reducing the amount of carbohydrates of a meal results in a lower GL of that meal, and therefore better glycemic control [24,25,26,27].

In a cross-over study of Stenvers et al. [26], 20 patients with T2D were randomly assigned to two groups. The control group consumed a free-choice breakfast, whereas the low-glycemic response (GR) group consumed an isoenergetic, low-GR meal replacement. In the low-GR meal replacement, carbohydrate content accounted for 42 E% compared to the 56 E% of the control meal, and fiber content accounted for 7 E%, approximately, compared to 1 E% for the control meal. The low-GR meal replacement resulted in significant reduction of postprandial glucose and insulin excursions. However, no differences were detected in fasting HbA1c after the completion of a 3 month intervention. The results of this study showed that replacing a meal with a low-GR meal substitute favors postprandial glucose excursions.

In a more recent randomized crossover trial, Chang et al. [25] investigated the effect of altered nutrient breakfasts on postprandial glucose response, but instead of altering GI through replacement of high-GI foods with low-GI ones, they altered GL by reducing the total amount of carbohydrate, resulting in a low-carbohydrate breakfast (LCBF). In the LCBF, carbohydrates provided approximately 10% of energy. It should be noted that participants were not following a low-carbohydrate diet, as their daily carbohydrate intake ranged from 180 to 270 g., nor a low-calorie diet, as their daily energy intake was approximately 2080 kcal. During the two trials of the study, participants consumed either the LCBF or a breakfast composed in accordance with the recommendations of dietary guidelines; therefore, carbohydrates contributed to approximately 55% of energy (GLBF). Lunch and dinner had the same macronutrient composition. Participants had a 74% lower glucose response post-breakfast when they consumed the low-carbohydrate breakfast, and 24 h iAUC was 32% ± 30% lower when compared with the GLBF group. Glycemic variability also improved significantly in the LCBF. No differences were detected in postprandial responses after lunch and dinner between the two groups.

Expanding the concept of the GL of a meal and with the aim to include all possible nutritional interventions that research suggests may reduce postprandial hyperglycemia, we included a recent cross-over clinical study of Imai et al. [24] that explored the immediate impact of a late-night dinner on postprandial glucose excursions. Researchers investigated whether the division of the late-night meal in two smaller meals consumed 3 h apart would instigate different glycemic responses. Participants were advised to consume three identical test meals (breakfast, lunch, dinner) for three consecutive days (from day 2 to day 4). On days 2 and 4, participants consumed either dinner at 21:00 h or the divided dinner at 18:00 h and 21:00 h. The divided dinner provided 70% of carbohydrates at 18:00 h and 30% of carbohydrates at 21:00 h. Division of the meal consequently resulted in two smaller meals with lower GL in comparison to the initial meal. The study showed that dividing the late-night meal led to a lower postprandial hyperglycemia, suggesting that it may be a practical and effective strategy for T2D patients to prevent diabetic complications.

Nutritional interventions focusing on the carbohydrate content of foods are not always simple to implement as they require nutrition counselling and education to enable patients with T2D to apply nutrition science in everyday life and adapt to new eating habits. Altering GL by increasing fiber content may be practical advice for patients with T2D.

Kamalpour et al. [27] divided patients with T2D into two groups: the MoCyllium group was consuming a higher carbohydrate diet (approximately 60% of total energy) with 7 g. of psyllium, and the LoCarb group was consuming a lower carbohydrate diet (approximately 50% of total energy) alone. Total calorie intake did not differ in both diets (energy in MoCyllium: 1399.19 ± 100.74, energy in LoCarb: 1397.76 ± 112.80). No statistical differences were observed regarding postprandial glucose in both groups. However, it is notable that insulin sensitivity increased in the MoCyllium group. Researchers concluded that patients who are unable to reduce carbohydrate amount in their diet should alternatively be advised to increase fiber content. 

In this analysis, 5 of 10 studies demonstrated that increasing soluble fiber intake to reduce GI of the meal has a positive effect on postprandial glucose in patients with T2D [22,27,28,29,30]. 

Silva et al. [28] investigated the effect of four different types of breakfasts in terms of GI content and amount of fiber in patients with T2D. Patients who consumed a high-GI/low-fiber breakfast had higher postprandial glucose concentrations compared with those who consumed a low-GI/high-fiber breakfast. It was also apparent that among the groups with the same GI, the presence of a higher fiber content resulted in a lower postprandial glucose response. However, because this study evaluated complete meals and not individual foods, it was not feasible to determine the most effective dietary strategy, that is, to increase fiber or to decrease the GI. 

De Carvalho et al. [30] conducted a study where participants consumed three isocaloric breakfasts with similar macronutrient content. They differed in the amount of fiber, resulting in meals with differing GI. Low-GI meals, with high fiber intake derived either from foods or supplements, were associated with lower postprandial glucose at breakfast. This study showed that increasing the amount of soluble fiber of a meal, either by high-fiber foods or by adding a supplement, may be a sound approach for improving the postprandial metabolic profile of patients with T2D.

Similarly, Lobos et al. [29] conducted a crossover clinical trial in patients with T2D under intensive insulin therapy (IIT) and tested two breakfast meals, a low-GI breakfast and a high-GI breakfast. The two meals were isocaloric and provided the same amount of macronutrients, differing only in the amount of fiber (fiber in low-GI: 4.9 ± 0.5, and fiber content in high-GI: 2.2 ± 0.2). Participants that consumed the low-GI breakfast presented a significant lower postprandial response in comparison to the group that consumed the high-GI breakfast (glucose area under the curve (AUC) was lower in low-GI breakfast with a difference of 46%).

**Table 2 nutrients-12-01561-t002:** Studies included in the review.

No.	Citation	Length of Study	Sample Size	Age (Years)	Treatment/Medication Type	Study Type	Intervention
1.	Wolever et al. [21]	12 months	162	35–75	Diet (*n* = 162)	Parallel design	Low-GI diet vs. higher GI
2.	Stenvers et al. [26]	22 months	20	30–75	N/A	Crossover	Low glucose response liquid formula vs. free choice control diet
3.	Goncalves Reis et al. [23]	2 weeks	12	40–75	Oral hyperglycemic agents (*n* = 8) Insulin (*n* = 2) Oral hyperglycemic agents with insulin (*n* = 2)	Crossover	Low-GI diet vs. higher GI
4.	Lobos et al. [29]	2 weeks	10	55 ± 6	Intensive insulin therapy (*n* = 10) Oral hyperglycemic agents (*n* = 7)	Crossover	Low-GI diet vs. higher GI
5.	Chang et al. [25]	3–4 days (wash-out ranged from 24 to 48h)	23	59 ± 11	Oral hyperglycemic agents (*n* = 15) Diet (*n* = 8)	Crossover	Reduced CHO content meal vs. higher CHO content meal
6.	Silva et al. [28]	32 days,4 intervention days separated by 7-day wash out)	14	65.8 ± 5.2	Oral hyperglycemic agents (*n* = 12) Diet (*n* = 2) Antihypertensive medication (*n* = 11) Lipid-lowering medication (*n* = 10)	Crossover	Combination of high-GI/high- or low- fiber vs. low-GI/high- or low-fiber meal
7.	Kamalpour et al. [27]	2 weeks	37	55 ± 1.2	Insulin release stimulators (*n* = 20) Metformin (*n* = 35) Acarbose (*n* = 3) Statins (*n* = 18) Aspirin (*n* = 11) Angiotensin converting enzyme (ACE-I) inhibitors/Angiotensin receptor blockers (ARB) (*n* = 14)	Parallel design	Moderate CHO meal + 7 g. Psyllium vs. lower CHO meal
8.	de Carvalho et al. [30]	24 days,3 intervention days separated by 7-day wash-out)	19	65.8 ± 7.3	Oral hyperglycemic agents (*n* = 18) Diet (*n* = 1) Antihypertensive (*n* = 16) Lipid-lowering (*n* = 16)	Crossover	Higher fiber diet (low-GI) meal vs. higher fiber supplement (low-GI meal) vs. usual fiber (higher GI meal)
9.	Nisak et al. [22]	12 weeks	41	55 ± 10	Oral hyperglycemic agents (*n* = N/A) Diet (*n* = N/A)	Crossover	Low-GI meal vs. higher GI meal
10.	Imai et al. [24]	5 days	16	N/A	Diet (*n* = 6) Oral hyperglycemic agents (*n* = 10)	Crossover	Higher CHO content meal vs. divided meal (resulting in lower CHO content meal)

GL-glycemic load; GI-glycemic index; CHO- carbohydrates.

**Table 3 nutrients-12-01561-t003:** Main findings and possible limitations of each study.

No.	Citation	Main Findings	Possible Limitations
1.	Wolever et al. [21]	Low-GI diet with low-GI meals resulted in a lower postprandial glucose response in patients with T2D compared to a low-GI diet with high-GI meals (*p* < 0.001)	Researchers did not compare the acute effects of the high-GI diet vs. low-GI diet and the high-GI diet vs. low-CHO diet in the same participants
2.	Stenvers et al. [26]	(i) Replacing breakfast with a low-GR meal substitute or (ii) reducing the amount of CHO and increasing the amount of fiber resulted in a reduced postprandial glucose response in T2D patients	No effect was observed in the fasting glucose and hemoglobin A1c (HbA1c), possibly due to the low baseline HbA1c values
3.	Goncalves Reis et al. [23]	Dietary advice on low-GI diet resulted in a significant reduction of CHO content of the diet, which led to a significantly lower postprandial glucose response on the first day	(i) At the end of the 3-day trial, no statistical significant differences were found (ii) Short duration of the intervention; applied for 3 days per week
4.	Lobos et al. [29]	A low-GI breakfast, as opposed to a high-GI breakfast, resulted in significant greater reduction of postprandial glucose response in patients with T2D under IIT (*p* < 0.022)	(i) Small sample size (ii) 90% of the sample corresponded to women, which decreases generalizability of the results (iii) Did not include measurements of insulin excursions
5.	Chang et al. [25]	(i) A low-CHO, high-fat breakfast where energy from CHO accounted for less than 10% energy lowered postprandial glucose excursions (*p* < 0.01); (ii) overall postprandial hyperglycemia and glycemic variability decreased with the low-carbohydrate diet (*p* < 0.03)	(i) Short duration of the intervention (ii) Research design did not include measurements of insulin excursions
6.	Silva et al. [28]	Plasma glucose, insulin, and ghrelin responses were least favorable for patients with T2D who consumed a high-GI and low-fiber diet (*p* < 0.005)	The tables of glycemic index were used to calculate GI instead of GI being determined from every food included in a meal
7.	Kamalpour et al. [27]	A low-calorie, reduced-CHO, and high-fiber diet was found to be beneficial and could improve fasting plasma insulin in poorly controlled T2D patients	No statistically significant differences observed regarding postprandial glucose and insulin levels, possibly due to sample size. Additionally, participants in the LoCarb group could not meet dietary advice, which resulted in smaller changes from baseline intakes
8.	de Carvalho et al. [30]	Increasing fiber content in breakfast, either from food or supplements, resulted in a lower postprandial glucose response (*p* < 0.05)	Insulin differences observed were not statistically significant. Additionally, participants had a good glycemic control at baseline, thus not permitting generalization of results
9.	Nisak et al. [22]	Postprandial glycemia and insulin responses were reduced after ingestion of a low-GI meal (*p* < 0.005)	Macronutrient content of the two test meals was not able to be kept identical, thus increasing possibility of confounding variables
10.	Imai et al. [24]	Dividing late night diner can significantly reduce postprandial hyperglycemia (*p* < 0.01)	The design of the study did not test the division of dinner in two meals with identical nutritional composition. The division resulted in a high-CHO meal, which was ingested first, and a high-protein, -fat, and -fiber meal, which was ingested approximately 3 h later; therefore, how consumption of the reverse nutritional composition would impact on postprandial hyperglycemia was left unknown

T2D-type 2 diabetes, GL-glycemic load; GI-glycemic index; CHO- carbohydrates.

## 4. Discussion

The present review aimed to examine the impact of diets with different GI and GL on hyperglycemia during the postprandial state. Several clinical studies on GI diets in human populations have been conducted, but research on acute blood fluctuations is limited by comparison. 

In patients with T2D, glucose homeostasis is impaired, and this impairment manifests in fasting and postprandial hyperglycemia. Specifically, postprandial hyperglycemia contributes greatly to the development of T2D complications because it contributes greatly to microvascular and macrovascular damage. Hyperglycemic spikes during postprandial state have been shown to instigate endothelial dysfunction, inflammation, and increase oxidative stress [31].

Postprandial hyperglycemia is therefore a major pathophysiological state that contributes to the development and further progression of micro- and macrovascular complications in T2D; thus, controlling postprandial hyperglycemia should be the focus of all nutritional interventions for T2D.

The analysis of the current review supports the findings of previous research regarding the positive effect of low-GI/GL meals on postprandial hyperglycemia in patients with T2D [32,33]. Postprandial glycemic response is primarily affected by carbohydrates; thus, the quality and quantity of carbohydrates are important determinants of postprandial glucose concentrations. Consequently, taking into consideration both the amount and type of carbohydrates consumed, the amount of soluble fiber [34,35] and GI of foods [32] are also key contributing factors to postprandial glucose responses.

Low-GI diets have been reported to improve the overall glycemic profile when used as intervention for patients with T2D, resulting in reduced HbA1c [1,13,36], but even when the overall glycemic profile appears to be good, postprandial hyperglycemia is still very frequent in patients with T2D [37]. Therefore, when specifically targeting to control postprandial glucose fluctuations, a low-GI/GL diet overall, without considering the carbohydrate type and amount of each meal, may be an inadequate strategy. To improve hyperglycemic spikes in the postprandial state throughout the day, it is important not to overlook the nutrient content of each meal separately.

In addition, breakfast composition seems to play a role in overall glycemic response because it is the meal that leads to the greatest postprandial hyperglycemic excursions [25,38]. Considering that postprandial glucose excursions are higher in the morning [8], it appears logical to focus on improving post-breakfast hyperglycemia, as it is a simple and practical strategy.

Many researchers have investigated the effect of different breakfasts on postprandial glucose in patients with T2D. Six of the studies included in this review investigated the effect of an altered breakfast in terms of GI/GL values, and all concluded that a low-GI/GL breakfast (either by reducing carbohydrates, increasing soluble fiber, or focusing on low-GI foods) can result in a lower postprandial glucose concentration [25,26,28,29,30,39]. This is in agreement with the findings of a previous study in participants with T2D [40], which also reported that breakfast meals of the same macronutrient content but different GI resulted in acute postprandial glucose responses that were proportional to their GI.

Because postprandial hyperinsulinemia related to postprandial hyperglycemia is implicated in the etiology of type 2 diabetes complications, it is important to investigate how low GI/GL affect insulin sensitivity. Among these six studies, only four included variables such as plasma insulin measurements and iAUC for insulin [26,28,30,39]. Apart from one study [30] that did not find significant differences in insulin levels, all three studies reported a beneficial effect. This is, however, inconsistent with the findings of two other studies of this review that reported no significant effect on insulin levels [21,27]. 

Generally, low-GI diets have a higher amount of fiber, which has been shown to further improve postprandial oscillations by lowering glycemic response immediately [23,41]. The beneficial effect of fiber, specifically soluble or viscous fiber, on postprandial glucose excursions has been reported in many previous studies [29,31,42]. Adding soluble viscous fiber in common foods increases dietary fiber content of the diet, as is recommended by Reynolds et al. in a recent a series of systematic reviews [42], and also lowers the GI of the food. 

Therefore, a low-GI meal or the addition of fiber in meals appear to be effective strategies to improve postprandial response in T2D patients [27,28,30]. These findings are in line with the results of the study conducted by Nisak et al. [22], in which low-GI/high-fiber meals led to improved postprandial hyperglycemia compared to a high-GI/low-fiber meal.

## 5. Strengths and Limitations of the Review

This review provides current evidence on the beneficial effect of low-GI/GL dietary interventions on postprandial hyperglycemia in patients with T2D. One limitation of the review is the relatively limited available number of studies that included measurements to assess postprandial glucose excursions and glucose variability in T2D. In addition, most studies had small sample sizes and short-term intervention. Another possible limitation is the inconsistent variables used to assess postprandial glucose responses (e.g., 2 h postprandial finger prick vs. 24 h iAUC).

## 6. Conclusions

At this time, a global strategy for a diet that addresses patients with T2D is yet to be approved. In consequence, research that will contribute to a better understanding of how the composition of meals affects postprandial glucose will in the future be of great importance for patients with T2D. 

Pursuing a low-GI diet, without considering the carbohydrate content and energy intakes in patients with type 2 diabetes, does not appear to be effective in managing PPG. Both the amount and the type of carbohydrate in the diet should be considered in conjunction when formulating dietary advice for patients with T2D. Following a low-GI diet is likely to lead to reduced carbohydrate consumption, which is well documented to favor patients with T2D as an effective way to improve postprandial glucose.

Significant reduction of carbohydrate content in breakfast is an effective way to improve postprandial glucose, even without changing the macronutrient content of the rest of the day’s meals. Educating patients on how to follow a low-GI diet can result in smaller carbohydrate intake that is well established to improve overall glycemic control. In poorly controlled patients, where nutrition education in macronutrient composition is not practical/feasible, advising to simply divide the latest meal of the day can result in better overall glycemic control. In addition, the idea of replacing meals, especially breakfast, with a meal substitute of prespecified macronutrients composition that will yield low glycemic responses may be an effective intervention when nutrition counselling and supervision is not feasible.

Either reducing the amount of carbohydrate in a meal or increasing consumption of soluble fiber has a favorable effect on postprandial glucose excursions. Adding a high soluble fiber supplement in meals leads to similar benefits with increased intake of high-fiber foods. This is a practical strategy to improve postprandial hyperglycemia in T2D. 

In conclusion, it is sensible to conclude that patients with T2D should be encouraged to reduce the carbohydrate content of a meal, that is, to consume a lower GL meal as well as increase soluble fiber content in meals. This research concludes with practical recommendations that are summarized in Table 4. 

## 7. Avenues for Future Research

In order to formulate useful nutritional advice based on the GI/GL target to improve the postprandial metabolic profile of patients with T2D, more studies need to be conducted that will incorporate larger size samples and a wide and consistent range of indices such as iAUC for glucose and insulin, as well as plasma concentration of glucose and insulin. Test meals used in such studies also need to match macronutrients to reduce confounding variables that make interpretation of the results difficult.

## Figures and Tables

**Table 1 nutrients-12-01561-t001:** Selection criteria based on population, intervention, comparator, and outcomes (PICO) framework.

PICO Framework	Inclusion Criteria	Exclusion Criteria
Population	Adults (18–75 years) with T2D	Studies involving patients with either T1D, gestational diabetes, animal studies
Intervention	(i) Low-GI/GL (ii) Reduced CHO (resulting in lower GL) (iii) Increased soluble fiber content (resulting in a lower GI)	Ketogenic diets, studies with no data of GI value of meals or no data on CHO content of meals
Comparator	Higher GI/GL diets and/or control	Studies using medical treatment as comparator
Outcomes	Postprandial glucose response	Studies not involving postprandial glucose measurements
Types of study	RCTs	Reviews, books, comments, qualitative studies

T1D-type 1 diabetes, T2D-type 2 diabetes, GL-glycemic load; GI-glycemic index; RCT-randomized control trials:CHO- carbohydrates.

**Table 4 nutrients-12-01561-t004:** Practical nutrition recommendations for improving postprandial hyperglycemia in type 2 diabetes (T2D).

▪ Reduce glycemic index (GI) of every meal by replacing high-GI foods with low-GI foods
▪ Try to reduce total reduced carbohydrate (CHO) content of diet and in every meal
▪ Increase daily intake of soluble fiber, aim to include soluble fiber in all meals
▪ Aim to have more than 2 g of soluble fiber in breakfast
▪ Reduce portion size of late-night dinner
▪ For breakfast, aim to have less than 50% of energy deriving from CHO

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
