# Peer review of "Glycemic Index (GI) or Glycemic Load (GL) and Dietary Interventions for Optimizing Postprandial Hyperglycemia in Patients with T2 Diabetes: A Review"

_nutrients, 2020, doi:10.3390/nu12061561_

Round 1

Reviewer 1 Report

This paper summarizes the literature on the effects of different nutritional strategies on the postprandial hyperglycemia and its importance in people with type 2 diabetes.

Minor Comments

  1. Introduction, spell check, correct “50 ram” to “50gram” (line number:71)
  2. Authors used abbreviation RCTs and did not mention what it is (line-113 and in TAble 1). Is it randomized controlled trial? Define abbreviation in the text.
  3. Authors did not mention age distribution/range for inclusion criteria. Please include in Table1.
  4. Under Table 2, add range of postprandial glucose levels for each group according to the studies listed. Authors also did not Include medication details of the subjects in the Table2 in relevant studies. (include if available)
  5. Authors are not consistent with usage of terms for Carbohydrate (CHO and carbs). Use any one of the abbreviation or word for carbohydrates, it would be nice to stick with “carbohydrate”. (Eg. CHO diet -line:153; carb diet-line:154 and carbohydrate diet-line:155; which is confusing for readers)
  6. In Table 3, authors mentioned “Avoid excess of CHO in breakfast” but did not define what is excess CHO, is there any threshold range of CHO content in the meal that should be considered? Please explain.
  7. Since Insulin level is also an important factor that contribute to postprandial blood glucose level, comment on Insulin levels after postprandial phase in relation to diet and glycemic load (GL).

Consider following publications

1: Am J Clin Nutr. 2015 Oct;102(4):801-6. doi: 10.3945/ajcn.115.112904. PMID: 26354547

2: Clin Nutr. 2019 Feb;38(1):465-471. doi: 10.1016/j.clnu.2017.11.010.PMID: 29248250

Author Response

1. Introduction, spell check, correct “50 ram” to “50gram” (line number:71)

Change has been made in the text please see line number 712.

2. Authors used abbreviation RCTs and did not mention what it is (line-113 and in TAble 1). Is it randomized controlled trial? Define abbreviation in the text.

Change has been made in the text please see line number 113

3. Authors did not mention age distribution/range for inclusion criteria. Please include in Table1.

Age range has now been included in table 1. Analytically for each study see also table 2.

4. Under Table 2, add range of postprandial glucose levels for each group according to the studies listed. Authors also did not Include medication details of the subjects in the Table2 in relevant studies. (include if available)

Medication has been included in table 2

5. Authors are not consistent with usage of terms for Carbohydrate (CHO and carbs). Use any one of the abbreviation or word for carbohydrates, it would be nice to stick with “carbohydrate”. (Eg. CHO diet -line:153; carb diet-line:154 and carbohydrate diet-line:155; which is confusing for readers)

The word carbohydrate has replaced all other terms. CHO abbreviation is only kept for tables and brackets

6. In Table 3, authors mentioned “Avoid excess of CHO in breakfast” but did not define what is excess CHO, is there any threshold range of CHO content in the meal that should be considered? Please explain.

In most studies that investigated alterations of GI in breakfast, it appears that a percentage of energy deriving from CHO that ranges between 20%-50% approx. was more beneficial than when percentage of energy from CHO exceed the upper level i.e. 50%.  Aiming for less than 50% of energy from CHO in breakfast is a safe, realistic, and practical advice for patients with T2D.

7. Since Insulin level is also an important factor that contribute to postprandial blood glucose level, comment on Insulin levels after postprandial phase in relation to diet and glycemic load (GL).

Postprandial insulin levels are included in the introduction (lines 107 -117) and in the discussion (lines 310-16)

Reviewer 2 Report

Overall

  • The authors interpreted and presented relevant results.
  • It presented a summary of the current state of understanding. The studies that were included are appropriate and published in the last 10 years.
  • The review was well written and easy to follow.
  • The topic is interesting and important.
  • Please replace the term subjects with participants.

Methods

  • It would be beneficial to report the number of titles, abstracts and full texts screened.

Results  

  • Although concise, not all 10 studies were included in the results section. References 23, 24 and 26 should be described in the results section since they were selected.
  • Also, the authors should have outlined the limitations of each study and how they may impact the results.
  • Reference 25. Stenvers et al. Please provide more details regarding the study design and results.
  • Reference 22. Chang et al. Include 24 h iAUC and glycemic variability results when discussing the study.
  • Reference 21. Imai et al. Need a more detailed description of the findings.
  • To address the above comments, a table outlining the main findings from each study could be added. This would be helpful to the reader and make it a more comprehensive review.

Discussion

  • The authors should have been more critical of the literature. What are the limitations of the current studies and gaps in the current literature?
  • What are the best avenues for future research?

References

  • Fix reference 26. Missing the journal name.

Author Response

Thank you very much for all your recommendations. 

Responses:

  • Please replace the term subjects with participants.

The term has been replaced as recommended.

Methods

  • It would be beneficial to report the number of titles, abstracts and full texts screened.

Number of full text, abstracts and titles articles are added.

Results 

  • Although concise, not all 10 studies were included in the results section. References 23, 24 and 26 should be described in the results section since they were selected.

Thank you for very much for your comments. All three references are now added in the result section. Reference 29 (lines 229-235), reference 23 (lines 178-190) and reference 22 (lines 167-175). References numbers have slightly change.

Ref 23- Lobos et al. àis now reference 29

Ref 24- Reis et al. àis now reference 23

Ref 26- Nisak et al. àis now reference 22

  • Also, the authors should have outlined the limitations of each study and how they may impact the results.

An additional table (Table 3) is added outlining the possible limitations of each study

  • Reference 25. Stenvers et al. Please provide more details regarding the study design and results.

Now reference 26. Thank you for your comment. The study design) is now added and more details were added regarding the results of the intervention (lines 196-204)

  • Reference 22. Chang et al. Include 24 h iAUC and glycemic variability results when discussing the study.

Now reference 25, Chang et al. 24h iAUC and glycemic variability are included (lines 205-218)

  • Reference 21. Imai et al. Need a more detailed description of the findings.

Now reference 24. A more detailed description is added (line 221 -227)

  • To address the above comments, a table outlining the main findings from each study could be added. This would be helpful to the reader and make it a more comprehensive review.

An additional table (Table 3) is added outlining the main findings

Discussion

  • The authors should have been more critical of the literature. What are the limitations of the current studies and gaps in the current literature?

Limitations are outlined in table 3. , also a section regarding strengths and limitations is added

  • What are the best avenues for future research?

A section regarding avenues for future research is included

References

  • Fix reference 26. Missing the journal name.

This reference is now reference number 22. Reference is now corrected and journal has been added.